# Physiological Cell Culture Media Tune Mitochondrial Bioenergetics and Drug Sensitivity in Cancer Cell Models

**DOI:** 10.3390/cancers14163917

**Published:** 2022-08-13

**Authors:** Omar Torres-Quesada, Carolina Doerrier, Sophie Strich, Erich Gnaiger, Eduard Stefan

**Affiliations:** 1Tyrolean Cancer Research Institute (TKFI), Innrain 66, 6020 Innsbruck, Austria; 2Institute of Biochemistry and Center for Molecular Biosciences, University of Innsbruck, Innrain 80/82, 6020 Innsbruck, Austria; 3Oroboros Instruments, Schoepfstrasse 18, 6020 Innsbruck, Austria

**Keywords:** cancer cells, kinase inhibitor, mitochondrial function, cell culture media, cell proliferation, cell bioenergetics

## Abstract

**Simple Summary:**

Cell biologists trust in standard media for analyzing cellular functions and for the specification of target-oriented drug efficacies in cell culture settings. Here, we present a general applicable workflow for the constant monitoring of bioenergetic states of cells grown in 2D cell models to accompany tailored drug discovery efforts. Using in-depth high-resolution respirometry analyses (HRR) of mitochondrial function, we unveiled that the human-plasma-like media (HPLM) altered cellular energetic states. In a systematic HRR setup for drug profiling experiments, we revealed an unexpected side effect of an FDA-approved cancer drug on mitochondrial function, exclusively in HPLM. Thus, we believe that both the recordings of bioenergetic states and the use of more physiological media would improve and reshape cell-based drug discovery ventures.

**Abstract:**

Two-dimensional cell cultures are established models in research for studying and perturbing cell-type specific functions. However, many limitations apply to the cell growth in a monolayer using standard cell culture media. Although they have been used for decades, their formulations do not mimic the composition of the human cell environment. In this study, we analyzed the impact of a newly formulated human plasma-like media (HPLM) on cell proliferation, mitochondrial bioenergetics, and alterations of drug efficacies using three distinct cancer cell lines. Using high-resolution respirometry, we observed that cells grown in HPLM displayed significantly altered mitochondrial bioenergetic profiles, particularly related to mitochondrial density and mild uncoupling of respiration. Furthermore, in contrast to standard media, the growth of cells in HPLM unveiled mitochondrial dysfunction upon exposure to the FDA-approved kinase inhibitor sunitinib. This seemingly context-dependent side effect of this drug highlights that the selection of the cell culture medium influences the assessment of cancer drug sensitivities. Thus, we suggest to prioritize media with a more physiological composition for analyzing bioenergetic profiles and to take it into account for assigning drug efficacies in the cell culture model of choice.

## 1. Introduction

Diverse cell culture models are commonly used to study the etiology and progression of a defined cancer cell population originating from patient specimens. Two-dimensional cell cultures (2D) provide standardized models suitable for reproducible drug screening and determination of drug efficacy by ensuring the accessibility of the experimental lead molecule to the cell monolayer. However, the artificial set-up for culturing cells is limited due to the lack of various physiological components, metabolites, and factors that are trackable, e.g., in the human plasma or in the tumor microenvironment [1]. It is thus an acute challenge to develop and advance media formulations with a more human-like (patho) physiological composition.

The selection of the growth medium influences the metabolism, and thus, cell growth in culture. In addition, the nutrient composition of the culture medium affects distinct cell-type-specific features of the cell [2]. This involves the exposure to diverse environmental stimuli for redirecting signaling events and gene expression signatures. Classical cell culture media should hold the promise to create the growth conditions for the in vitro cell lines. However, usually, these formulations do not mimic the physiological conditions or the composition of the tumor microenvironment. Among the most frequently used cell culture media are Eagle’s Minimal Essential Medium (MEM), its Dulbecco’s modified version (DMEM), and RPMI 1640 [3,4,5]. They were designed to support the continuous growth of cancer cells. However, the composition and concentrations of several metabolites, including glucose and amino acids, do not reflect the molecular composition of human plasma. Recently, two new formulations of cell culture media have been developed; Human Plasma-like Medium (HPLM, Thermofisher, Waltham, MA, USA) and the similar Plasmax^TM^ (Ximbio, London, UK) were established as alternatives to classical cell culture media [6,7]. HPLM contains a cocktail of 31 physiological components that are absent in commonly used cell culture media. Furthermore, HPLM contains physiologically relevant concentrations of a collection of typical media components such as glucose, amino acids, and electrolytes [6]. Importantly, cells grown in HPLM compared to classical media formulations have different cellular signatures that are related to adaptions of the metabolome, redox state, de-novo pyrimidine synthesis pathway, and lipid peroxidation and ferroptosis [6,7,8]. HPLM induces changes in the gene expression of human essential genes [9].

Mitochondrial metabolism is linked to different stages of carcinogenesis [10,11,12]. Mitochondria act as dynamic cell organelles by orchestrating the cellular energy transformation. Besides being the bioenergetic powerhouse of the cell, mitochondria play a crucial role in many central cellular processes such as biosynthesis, calcium and lipid homeostasis, cell division, and apoptosis [13,14]. The metabolic reprogramming of cancer cells is sustained by the accessibility of nutrients and secondary intracellular alterations and adaptions of signaling events. Indeed, environmental conditions in cell culture models impact the potencies and efficacies of intervention strategies [15,16,17]. This variability of ill-defined cell culture conditions may contribute to discrepancies of in vivo and in vitro studies focusing on mitochondrial bioenergetics and off-target drug effects [18,19,20].

Here, we set out to determine the precise impact of cell culture media compositions on cell proliferation and mitochondrial function. To determine the mitochondrial bioenergetic profiles of several standard cancer cell lines, we used high-resolution respirometry (HRR) for the quantitative evaluation of cell respiration [21]. Furthermore, we evaluated whether HPLM influences mitochondrial drug sensitivity in cells exposed to the tyrosine kinase inhibitor sunitinib. The newly formulated HPLM had distinct effects on cell proliferation and mitochondrial respiration under steady-state conditions and after exposure to sunitinib when compared to classical cell culture media. Overall, this study highlights the necessity to evaluate the choice of cell culture media for studying pharmacological targeting of key events of metabolic and pathophysiological functions.

## 2. Materials and Methods

### 2.1. Cell Culture

The human cancer cell lines were purchased from ATCC (Manassas, VA, USA): SW620 (ATCC^®^ CCL-227™, metastatic-derived colon cancer), the MCF7 (ATCC^®^ HTB-22™, breast cancer,) and A375 (ATCC^®^ CRL-1619™, malignant melanoma). Cells were incubated at 37 °C in a humidified incubator with 5% CO_2_ in air. For the SW620 and MCF7 cell lines, we generated a RPMI formulation with physiological glucose concentration (5 mM) supplemented with 10% dialyzed fetal bovine serum (FBS) (PAN BIOTECH, Aidenback, Germany, P30-2102) to reduce the interference with serum metabolites. For the A375, we generated a DMEM with 5 mM glucose supplemented with 10% dialyzed FBS. The new HPLM (Thermo Fisher Scientific, A4899101) was supplemented with 10% dialyzed FBS. Cells were counted in a Cellometer^®^ counter (Nexcellon Bioscience, Lawrence, MA, USA) following the manufacturer’s instructions. Briefly, a 100 µL aliquot of the cells was diluted in 0.1% trypan blue and measured in a Cellometer^®^ chamber.

For the sunitinib treatments, SW620 cells were cultured either in RPMI or HPLM supplemented with 10% dialyzed FBS. At the day of the experiment, cells were treated for 3 h with 10 µM sunitinib (MedChemExpress, Monmouth Junction, NJ, USA HY-10255A) dissolved in DMSO. After the incubation period, the cells were washed with PBS, detached with trypsin EDTA, counted, and subjected to respirometry.

### 2.2. Proliferation Assays

For the proliferation studies, cells were seeded in a 96-well plate format (Corning) with a density of 10,000 cells/well. Cell growth was monitored during 100 h with scanning points every 4 h in an Incucyte^®^ S3 Live-Cell Analysis System (Essen Bioscience, Royston, United Kingdom). Phase-area confluency was calculated using the Incucyte^®^ analysis software. Area-under-the curve (AUC) and marginal distribution (slope variation) were performed using the software GraphPad Prims 6.

### 2.3. Sample Preparation and High-Resolution Respirometry

Prior to the respirometric measurements, SW620 and MCF7 cell populations were grown for four days in RPMI^+FBS^ and HPLM^+FBS^; A375 cells were grown in DMEM^+FBS^ and HPLM^+FBS^. Next, cells were trypsinized and resuspended in the corresponding cell culture media at a cell concentration of approximately 1.0 × 10^6^ x/mL for SW620 and A375 and 0.7 × 10^6^ x/mL for MCF7. In case of cells exposed to sunitinib, 1.0 × 10^6^ x/mL SW620 cells were suspended in the cell media containing the same concentration of solvent or inhibitor used for the treatments. The elementary unit “x” represents the single individual cell [21].

O_2_ consumption in living cells was measured using the O2k-FluoRespirometer (O2k, Oroboros Instruments, Innsbruck, Austria), a modular system for HRR. Temperature was kept constant at 37 °C ± 0.002 °C (electronic Peltier temperature control) under constant stirring at 750 rpm, which ensured a homogenous O_2_ concentration in the experimental closed chambers with experimental volumes of 2 mL. O_2_ concentration *c*_O2_ [μM] and O_2_ flow per cell *I*_O2_ [amol × s^−1^ × x^−1^] [21] were monitored real-time using DatLab 7.4 software (Oroboros Instruments, Innsbruck, Austria). Calibrations of the polarographic O_2_ sensors of the O2k were performed daily using an O_2_ solubility factor relative to pure water of 0.89 as in serum [22] and correcting the partial O_2_ pressure for barometric pressure monitored by the O2k. Instrumental O_2_ background tests were performed at monthly intervals [23]. The coupling control protocol for living cells (SUIT-003 D009) was applied; it allows the study of the different coupling control states in living cells: ROUTINE, LEAK and electron-transfer pathway capacity [21]. Cell addition into the O2k chambers was performed by complete volume replacement of the sample. ROUTINE respiration *R* is a physiological coupling state in which respiration is controlled by aerobic ATP demand and coupling efficiency. This step was followed by the inhibition of the phosphorylation system with oligomycin Omy, which induces LEAK respiration *L*. The inhibitory concentration of 10 nM was tested for each cell line. Subsequently, the protonmotive force was dissipated by stepwise titration (0.5 μM/step) of the uncoupler U carbonyl cyanide 3-chlorophenylhydrazone (CCCP) until the maximum noncoupled respiration was obtained (ET capacity *E*). The addition of 0.5 μM rotenone (Rot) and 5 μM antimycin A (Ama) inhibited the ET pathway, allowing the measurement of residual oxygen consumption *Rox*. Data analysis was performed using DatLab 7.4. Respiratory rates were corrected for the instrumental O_2_ background flux, dilution of the sample by titrations, and *Rox*.

To provide respiratory control fingerprints of coupling control, flux control ratios *FCR*, coupling-control ratios, and flux control efficiencies were calculated. *FCR* provide normalizations of O_2_ flow in any respiratory state (*i*) of a SUIT protocol by an internal functional mitochondrial marker. *FCR* are independent of cell count, cell size, and mitochondrial density [21],
$$FCR=\frac{I_{O_{2}}\left(\mathrm{state}\;i\right)}{I_{O_{2}}\left(\mathrm{reference}\;\mathrm{state}\right)}$$

ET capacity *E* was measured as the maximum respiration obtained in uncoupler titrations and defined as the common reference state for flux control ratio calculations (*E*/*E* = 1). Additionally, *L*/*R* coupling-control ratios, *E*-*L* net ET capacities, net *R*/*E* control ratios, *E*-*L* coupling efficiencies, and *R*-*L* coupling efficiencies were calculated [21,24].

### 2.4. Statistical Analysis

Statistical significance levels *p* were assessed using the non-parametric unpaired Mann–Whitney *t*-test. For multiple comparisons (respiratory control coupling states), a two-way ANOVA with Bonferroni’s multiple comparison test was used. Statistical analyses were performed using the software GraphPad Prims 6.

## 3. Results

### 3.1. HPLM Influences Steady State Cancer Cell Proliferation and Mitochondrial Function

To analyze the influence of more physiological cell culture media on key aspects of cancer cell metabolism, we selected several hugely proliferative human cancer cell lines. All of them are standard cell culture models, originating from different human tissues, displaying different phenotypic features (different cell size and morphology) and a different mutational burden.

To assess the impact of HPLM on key features of the SW620, MCF7, and A375 cancer cell lines, we compared the proliferation profiles of cell populations cultured in classical media and the new HPLM (denoted as RPMI^+FBS^, DMEM^+FBS^ and HPLM^+FBS^). We used dialyzed FBS to get a more defined environment of small molecules, since dialysis reduces the concentration of low molecular weight components of the serum (e.g., metabolites, nucleotides, amino acids). The differences of formulations between RPMI, DMEM, and HPLM are summarized in Figure 1.

First, we tracked cell proliferation kinetics in terms of confluence as a function of time and the proliferation rates as the area under the curve AUC. The linear regressions of these data sets highlight differences of growth curves for two cell lines. The proliferation rate in SW620 cells cultured in HPLM^+FBS^ was decreased, starting at 40 h and causing a difference of 29.5 h. This led to a strong delay of SW620 growth in HPLM^+FBS^ (Figure 2a–c). Differences in MCF7 proliferation were undetectable (Figure 2d–f). In contrast, the proliferation rates were augmented in the melanoma cell line A375 in the presence of HPLM^+FBS^ (Figure 2g–i). Taken together, HPLM supported the growth of all three cell lines. However, HPLM^+FBS^ surprisingly exerted opposite effects on cancer cell proliferation of SW620 and A375 cell lines.

To address whether HPLM^+FBS^ affects mitochondrial respiratory control in these cell lines, we measured cell respiration by HRR. Results on cell respiration of living cells are shown in Figure 3. (1) HPLM^+FBS^ exerted no detectable effects on the cell respiration of the SW620 cell line in the ROUTINE, LEAK, and ET states (Figure 3a,b). However, a higher concentration of the uncoupler CCCP was needed to reach the maximum ET capacity in cells grown in HPLM^+FBS^ compared to RPMI^+FBS^ ([U] = 7.0 ± 1.1 µM for RPMI^+FBS^ versus [U] = 9.5 ± 1.6 µM for HPLM^+FBS^; Figure 3a), which can be related to different uncoupler kinetics in HPLM (i.e., mitochondrial membrane stability); (2) ROUTINE respiration and to a larger extend ET capacity were increased in MCF7 cells grown in HPLM^+FBS^ (Figure 3c,d). (3) Respiration in all coupling control states—with major effects on ROUTINE respiration and ET capacity—was increased in A375 melanoma cells cultured in HPLM^+FBS^ compared to DMEM^+FBS^ (Figure 3e,f). Results are summarized in Appendix A.

To determine the coupling control in these cells, we calculated respiratory flux control ratios (*FCR*), coupling-control ratios (*L*/*E*, *R*/*E* and *L*/*R*), net respiratory capacities (*E*-*L* net ET capacity and net *R*/*E* ratio), and flux control efficiencies (*E*-*L* and *R*-*L*) (Figure 3g–i; summarized in Appendix A). *FCR* are O_2_ flux ratios normalized for the flux in a reference state, giving a fingerprint of mitochondrial coupling control independent of changes in mitochondrial content, cell volume, and sample amount [21,23,25]. Coupling control ratios are *FCR* at a constant mitochondrial pathway state. In the case of living cells, the coupling control ratios *R*/*E* and *L*/*E* are *FCR* defined by the upper and lower limits 1.0 to 0.0, where the reference state is the ET capacity with maximum respiratory rate. The ratio *E*/*E* is the normalization of the *E* fluxes against the ET capacity reference state, which relies in *FCR* of 1 [21]. In the SW620 cell line, the *FCR* for ROUTINE respiration (*R*/*E*, Figure 3g) was slightly increased in HPLM^+FBS^. Furthermore, we observed in SW620 cells grown in HPLM^+FBS^ a significant reduction of the *L*/*R* coupling-control ratio and elevation of the net *R*/*E* control ratio and *R*-*L* control efficiency (Appendix A) [26]. ATP demand and uncoupling or dyscoupling are drivers of ROUTINE respiration in living cells. Consequently, higher *R*/*E* may indicate an increase in uncoupling. However, no differences were observed between both media in LEAK respiration (Figure 3b; Appendix A), *L*/*E* coupling-control ratio (Figure 3g; Appendix A), and *E*-*L* coupling efficiency (Appendix A). Therefore, our findings might reflect that the increase in *R*/*E* ratio and *R*-*L* control efficiency in cells grown in HPLM^+FBS^ could be related to a shift from glycolytic to oxidative phosphorylation increasing *R* in HPLM^+FBS^ compared to RPMI^+FBS^ [21]. Additionally, no differences were observed between both media (Appendix A) when applying bioenergetic cluster analysis (BCA) [24,27], indicating that the mitochondrial bioenergetic profiles are similar. Second, in MCF7 cells, higher *L*/*E*
*FCR* (Figure 3h; Appendix A) and lower *E*-*L* coupling efficiency (Appendix A) in RPMI^+FBS^ compared to HPLM^+FBS^ indicate higher intrinsic uncoupling (physiological not-fully coupled mitochondria) in cells grown in RPMI^+FBS^. BCA in MCF7 cells identified two clusters in the scatter when LEAK respiration is expressed over ROUTINE respiration (Figure 4a), and ROUTINE respiration (Figure 4b), LEAK respiration (Figure 4c), and *E*-*L* coupling efficiency (Figure 4d) are expressed over ET capacity. BCA supports that respiration in MCF7 cells is better coupled when cells are grown in HPLM^+FBS^, by showing a drop in *E*-*L* coupling efficiency over ET capacity in the cluster RPMI^+FBS^ compared to HPLM^+FBS^ (Figure 4d). This lower coupling control efficiency in RPMI^+FBS^ might be caused by an ET- and uncoupling-linked effect observed in RPMI^+FBS^ (Figure 3d,h; Appendix A) [24,27]. Finally, we applied the same analysis to data sets of A375 cells. The *FCR* showed no difference between DMEM^+FBS^ and HPLM^+FBS^ (Figure 3i), indicating that the changes in O_2_ fluxes might originate from an increase in mitochondrial content and/or density. BCA analysis supports the same observation, with the generation of two isolinear clusters with similar coupling efficiencies (Figure 4e–h). Interestingly, the A375 cell line showed a reduction of *Rox* in cells grown in HPLM^+FBS^ as compared with DMEM^+FBS^ (Appendix A).

### 3.2. HPLM Modifies Mitochondrial Drug Sensitivity in Cancer Cells

After the characterization of mitochondria in steady-state conditions, we asked whether HPLM influences drug sensitivity in cancer cells. For this purpose, we cultivated the SW620 cells either in RPMI^+FBS^ or HPLM^+FBS^ and then exposed them to the receptor tyrosine kinase inhibitor sunitinib (Suten, SU11248), which is currently used as a cancer therapy against renal cell carcinoma and gastrointestinal stromal tumor [28,29]. Following treatments with sunitinib, we performed respirometric measurements. First, lower uncoupler concentrations were needed to reach ET capacity *E* in cells grown in HPLM^+FBS^ and treated with sunitinib (HPLM DMSO: [U] = 11.2 ± 4.5 µM versus HPLM sunitinib: 9.4 ± 5.0 µM) in contrast to cells grown in RPMI^+FBS^ (RPMI DMSO: [U] = 5.8 ± 1.5 µM vs RPMI sunitinib: 5.9 ± 1.8 µM, Figure 5a,b). Second, in contrast to cells grown in RPMI^+FBS^, SW620 cells exposed to sunitinib and grown in HPLM^+FBS^ showed a decrease in the ET capacity (Figure 5c,d; Appendix A).

Third, *R*/*E* and *L*/*E* ratios were elevated in cells grown in HPLM^+FBS^ and exposed to sunitinib as compared with DMSO (Figure 5f; Appendix A). This effect was not observed in RPMI (Figure 5e). Taken together, these results suggest that sunitinib exerts an inhibitory effect on the electron transfer system and additionally a partial dyscoupling effect (extrinsic experimentally-induced dyscoupling, i.e., pharmacological, pathological [30]), as indicated by the increase of *L* and *L*/*E*. Although dyscoupling reduces the efficiency of ATP production, this is in part compensated for by an increase in ROUTINE respiration. These conclusions are corroborated by the drop in *E*-*L* net ET capacity and *E*-*L* coupling efficiency in cells exposed to sunitinib, whereas the net *R*/*E* control ratio was constant (Appendix A). The slight decrease in the *R*-*L* control efficiency (*R*-*L*)/*R* (Appendix A) reflects a drop in the fraction of ROUTINE respiration coupled to phosphorylation, indicating that mitochondrial dyscoupling by sunitinib when cells are grown in HPLM^+FBS^ was not fully compensated for by an increase of ROUTINE respiration [24,27,31]. Bioenergetic cluster analysis (BCA) extends these observations (Figure 6a–d), showing two heterolinear clusters when LEAK respiration was expressed over ET with a cluster of cells grown in HPLM^+FBS^ treated with sunitinib with higher *L* (Figure 6c). Furthermore, two clusters were detected with a drop in the coupling efficiency in cells grown in HPLM^+FBS^ and treated with sunitinib (Figure 6d). These differences were not observed in cells grown in RPMI^+FBS^ (Appendix A). We did not observe any effect of sunitinib treatment neither in cell viability nor apoptosis in our experimental conditions (Appendix A).

Finally, we present evidence that the cancer drug sunitinib causes mitochondrial dysfunction only in SW620 cells grown in HPLM, whereas this effect was masked in cells grown in a classical culture medium. In conclusion, drug efficacies and off-target effects are cell type-specific and dependent on the culture media composition, and mitochondria can be used as an indicator for drug off-target effects.

## 4. Discussion

Cancer drug discovery in the preclinical phase starts in many cases with cell culture models. In the first-stage of cell-based drug evaluations, cell culture experiments are initiated to determine lead molecule potencies, efficacies, and the toxicity profile [32,33,34,35,36,37]. One major drawback of drug discovery is the subsequent translation to animal models, which depend on the reliability of these preclinical evaluations. The lack of reproducibility and the discrepancies between in vivo and in vitro models call for better defined cell culture conditions, especially for analyzing cancer pathophysiology. Recent studies demonstrate that the culture medium formulation significantly affects the results obtained in commonly used cell biology assays (i.e., colony formation, uridine metabolism, transcriptomics) [6,7,8,38].

The correlation of cancer cell proliferation and mitochondrial function goes back to the work of Otto Warburg. The ‘Warburg effect’ describes that high glycolytic fluxes are accompanied with defects of mitochondria in cancer cells [39]. The explanation evolved over decades, highlighting that some cancer cell types boost the function of the energetic organelle to promote cell proliferation [40]. In some cases, elevated OXPHOS levels may not be required to fulfill high proliferation rates. Cancer cells showing elevated glycolytic fluxes might have high proliferation rates showing low levels of aerobic cell respiration (i.e., liver and kidney cancer) [41,42,43]. In other cases, elevated OXPHOS levels can be supported by changes in mitochondrial dynamics during cell proliferation (e.g., fission, fusion, biogenesis). One example is that mitochondrial fusion supports enhanced levels of OXPHOS in transformed and non-transformed mouse fibroblasts [44].

In our study we evaluated more physiological cell culture conditions, deciphering the impact of the newly formulated HPLM on proliferation and cell respiration and in three standard human cancer cell lines. Moreover, we tested whether the effect of the FDA-approved multikinase inhibitor sunitinib on mitochondrial function is influenced by growth conditions in HPLM. First, HPLM supported the cell growth of all cancer cell lines tested. Second, HPLM exerts in a cell-type-specific manner diverse effects on cell proliferation when compared to the classical media. Our data are consistent with previous observations showing that physiological media formulation affects proliferation kinetics in cancer cells [6,7]. Third, HPLM altered the mitochondrial bioenergetic profiles in two of the three cell lines in a cell type-specific manner. Fourth, HPLM revealed a mitochondrial dysfunction produced by cells exposed to the cancer drug sunitinib that was not present in the respective cells grown in classical media. Our results provide a fingerprint of bioenergetic profiles in living cells for analysis of the effects of HPLM in steady state and under treatment with kinase inhibitors. We successfully implemented respiratory protocols with coupling-control ratios, flux control efficiencies, and bioenergetic cluster analysis and provide a general applicable workflow for continuously monitoring mitochondrial bioenergetic states for 2D cell culture drug discovery efforts.

In our experiments, cell respiration was improved in two of the three cell lines. However, cell respiration of cancer cell lines was differentially affected. In SW620 cells, respiratory rates were not affected but changes in coupling-control ratios related to ROUTINE respiration indicate that HPLM influences energy demand (increase of *R*/*E* with constant *E* and without uncoupling/dyscoupling). This supports the idea that metabolism switches to more aerobic metabolism in SW620 cells and leads to reduced proliferation rates in HPLM, which might be related to the glycolytic dependence of colon cancer cells. In MCF7 and A375 cells, we observed an increase of ROUTINE respiration and ET capacity that was more pronounced in the melanoma cell line. In MCF7 cells, this enhancement related to a higher coupling efficiency results in a higher mitochondrial quality with HPLM, indicating that the mitochondria are more efficient in oxidizing substrates without affecting cell proliferation [45]. In A375 cells, HPLM increased respiration in all coupling control states, suggesting that HPLM enhanced mitochondrial biogenesis without alteration of coupling efficiency. Our results suggest that a correlation of increased in mitochondrial respiration and cell proliferation might be the case for A375 cells (increasing on mitochondrial density and proliferation rate in HPLM). Furthermore, our respiratory studies do not exclude the possibility that HPLM might affect the aerobic glycolysis phenotype that is found in many cancer cell lines [46]. This mechanism might be relevant in connection with our findings that sunitinib exerts a mitochondrial dyscoupling effect in SW620 when grown in HPLM. This would make the cells more sensitive to the drug due to the high dependence on the glucose metabolism and glucose uptake observed in colon cancer cells [47,48,49]. Our findings support previous observations that the cell environment dictates cancer drug sensitivity, e.g., for the mitochondrial drug metformin [50,51]. This underlines the fact that off-target toxicity driven by drugs usually produces a mitochondrial dysfunction related to the coupling state [18,52,53].

## 5. Conclusions

The results presented in this work show that HPLM alters both the proliferative behavior and mitochondrial bioenergetic states of living cells when compared to classic cell culture media. These observations obtained with three highly proliferative cancer cell lines of different origin need to be considered in future studies related to molecular analyses and using intervention strategies versus cancer cell models. HPLM mimics a more physiological situation leading to differences in cell proliferation and cell respiration that needs to be considered for drawing translational conclusions. Importantly, one FDA-approved drug impaired mitochondrial function depending on the media composition. This observation must be considered for interpretations of drug efficacies and their off-target effects. Further standardization of cell culture media will help to solve the current reproducibility crisis in cancer biology [54] and, in general, in in vitro 2D cell models. Our study should increase the awareness of the importance of the cell culture setup to better streamline the time and cost-intensive quest for identifying more effective lead molecules in preclinical cancer research.

## Figures and Tables

**Figure 1 cancers-14-03917-f001:**
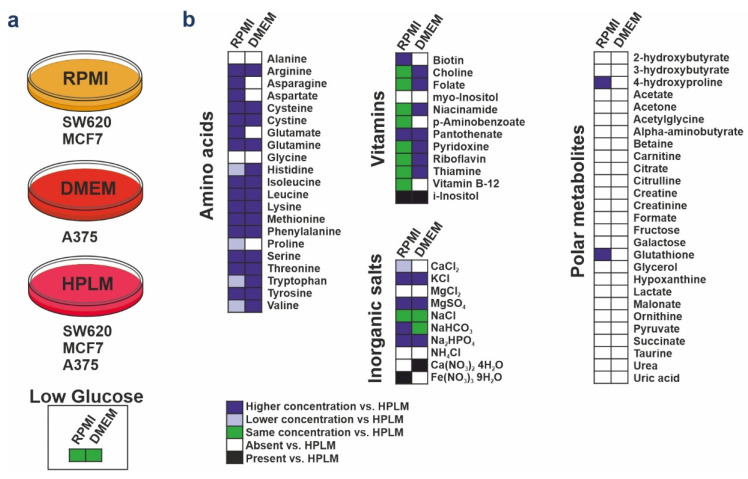
Media formulation of RPMI, DMEM, and HPLM. (**a**) All experiments were performed in these media supplemented by 10% dialyzed FBS. Glucose concentration of DMEM and RPMI was modified as physiological concentration (low glucose, 5 mM). (**b**) The chart shows the relative concentrations of RPMI and DMEM media supplemented with 5 mM glucose compared with HPLM. Components not present in RPMI and DMEM but present in HPLM are denoted as “absent”; components not present in the HPLM but present in the other media are denoted as “present”.

**Figure 2 cancers-14-03917-f002:**
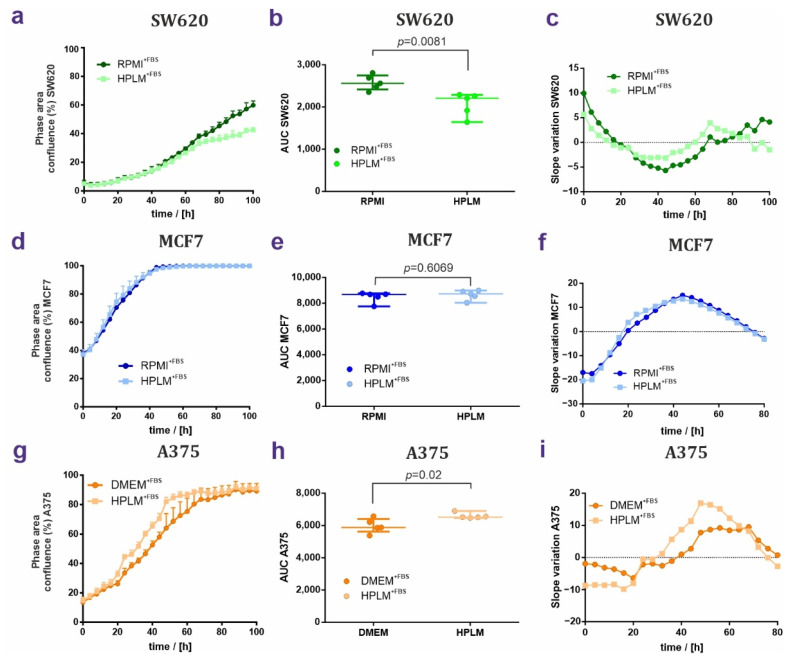
HPLM and RPMI/DMEM effects on cell growth of SW620, MCF7, and A375 cells. (**a**–**c**) Proliferation of SW620 cells grown in RPMI^+FBS^ or HPLM^+FBS^. Growth curves are shown in (**a**), AUC are plotted in (**b**), and slope variations are shown in (**c**). (**d**–**f**) Same analyses performed for MCF7 cells grown in either RPMI or HPLM showing proliferation curves (**g**), AUC (**h**), and slope variation (**i**). (**g**–**i**) Proliferation of the A375 cell line grown either in DMEM^+FBS^ or HPLM^+FBS^. In proliferation curves and AUC, values are represented as median ± IQR (50% range). Unpaired non-parametric *t*-test analysis; *N* = 5.

**Figure 3 cancers-14-03917-f003:**
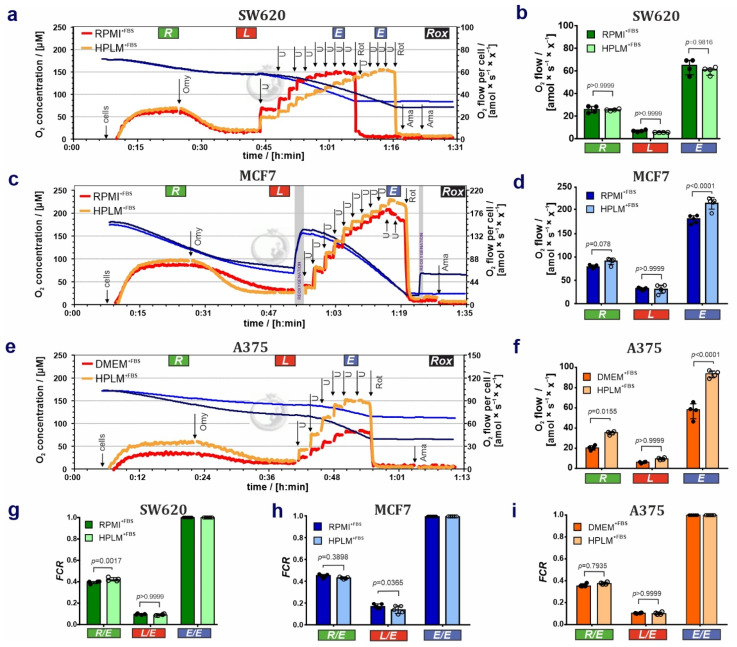
HPLM alters mitochondrial function in SW620, MCF7, and A375 cell populations when compared to standard cell culture media. (**a**,**c**,**e**) Representative respiratory traces for coupling control protocol SUIT-003-D009 in (**a**) SW620, (**c**) MCF7, and (**e**) A375 cells. Blue lines: O_2_ concentration [µM]; dark orange lines: O_2_ flow per cell [amol × s^−1^ × x^−1^] in cells grown and measured in (**a**,**c**) RPMI^+FBS^ or (**e**) DMEM^+FBS^; light orange lines: O_2_ flow per cell [amol × s^−1^ × x^−1^] in cells grown and measured in HPLM^+FBS^. “x” represents the unit cell. Sequential steps: cell addition, oligomycin (Omy), uncoupler CCCP (U), rotenone (Rot) and antimycin A (Ama). Total ROUTINE respiration (*R*’_tot_), LEAK respiration (*L*’_tot_), and ET capacity (*E*’_tot_) not baseline-corrected for residual O_2_ consumption (*Rox*). (**b**,**d**,**f**) O_2_ flow (*R*, *L*, *E*) amol × s^−1^ × x^−1^ ] baseline-corrected for *Rox* in (**b**) SW620, (**d**) MCF7 and (**f**) A375 cell lines. (**g**–**i**) Flux control ratios *FCR* in SW620, MCF7, and A375 cells lines. Results are represented as median ± IQR (50% range). 2-way ANOVA with Bonferroni’s multiple comparison test or non-parametric unpaired *t*-test analysis; *N* = 4–5.

**Figure 4 cancers-14-03917-f004:**
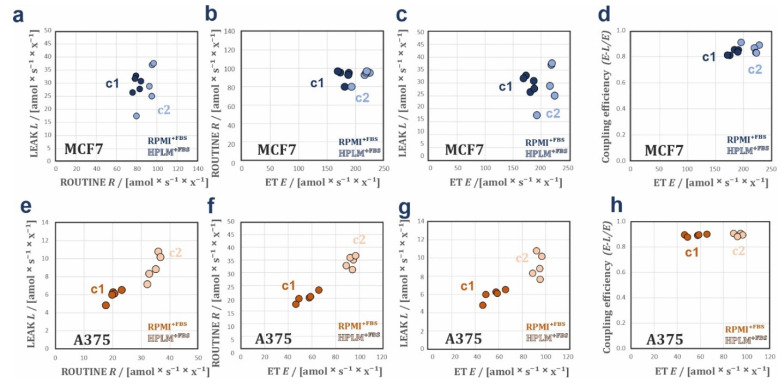
Bioenergetic cluster analyses in different respiratory coupling states of the SW620, MCF7, and A375 cell lines. BCA plots of steady state conditions (**a**–**h**). Plots of *L* over *R*, *R* over *E*, and *L* over *E*. (**a**–**c**) MCF7 showing two heterolinear bioenergetic clusters in RPMI^+FBS^ (c1) and HPLM^+FBS^ (c2). This indicates changes in mitochondrial quality between both media. (**e**–**g**) A375 showing two isolinear bioenergetic clusters in DMEM^+FBS^ (c1) and HPLM^+FBS^ (c2). Proportionality with zero intercept indicates changes of mitochondrial density in the cells grown in the two media. (**d**,**h**) Coupling efficiencies (*E*-*L*)/*E* over ET capacities in, MCF7 (**d**) and A375 (**h**). The lower (*E*-*L*)/*E* values in MCF7 cells in RPMI^+FBS^ at lower ET values indicate a drop in mitochondrial quality with RPMI^+FBS^. In A375, the (*E*-*L*)/*E* values remain constant over ET in both media indicating that mitochondrial quality is preserved.

**Figure 5 cancers-14-03917-f005:**
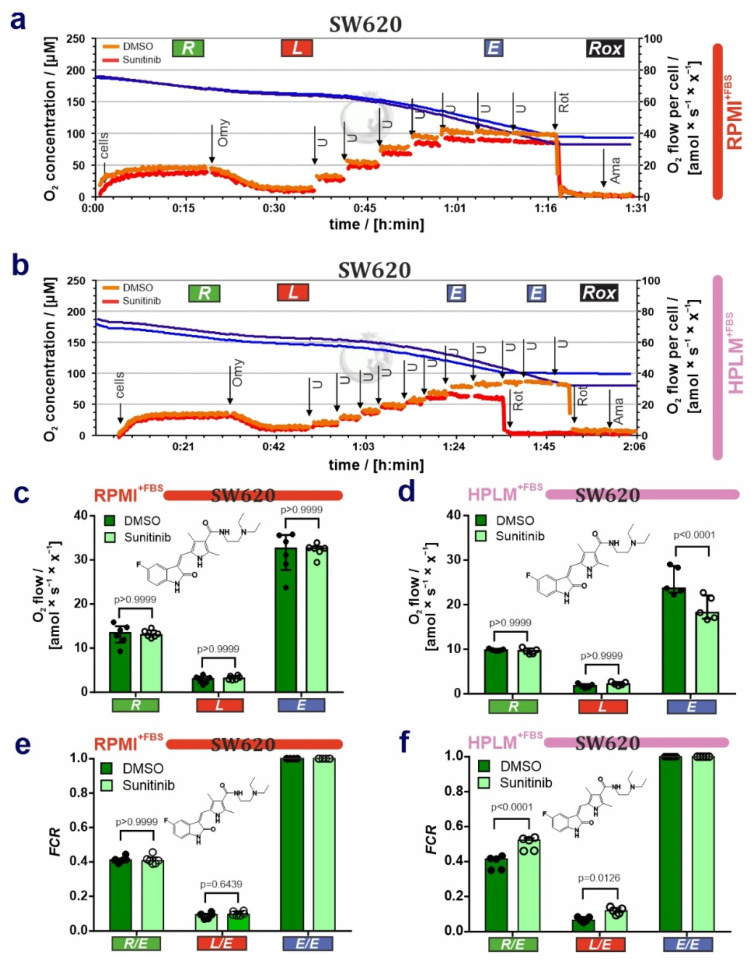
Differential effects of sunitinib on cell respiration of SW620 cells grown in RPMI or HPLM. (**a**,**b**) Representative respiratory traces for coupling control protocol SUIT-003-D009 in SW620 cells grown in (**a**) RPMI^+FBS^ and (**b**) HPLM^+FBS^, and exposed to 10 µM sunitinib for 3 h. Blue lines: O_2_ concentration [µM]; light orange lines: O_2_ flow per cell amol × s^−1^ × x^−1^ ] in DMSO control; and dark orange lines: O_2_ flow per cell [amol × s^−1^ × x^−1^] in sunitinib-treated cells. “x” represents the unit cell. (**c**,**d**) O_2_ flow (*R*, *L*, *E*) [amol × s^−1^ × x^−1^] baseline-corrected for *Rox* in (**c**) RPMI^+FBS^ and (**d**) HPLM^+FBS^. (**e**,**f**) Flux control ratios *FCR* in SW620 cells grown in (**e**) RPMI^+FBS^ and (**f**) HPLM^+FBS^ and exposed to sunitinib (chemical structure depicted). Median ± IQR (50% range). 2-way ANOVA with Bonferroni´s multiple comparison test; *N* = 5–6.

**Figure 6 cancers-14-03917-f006:**
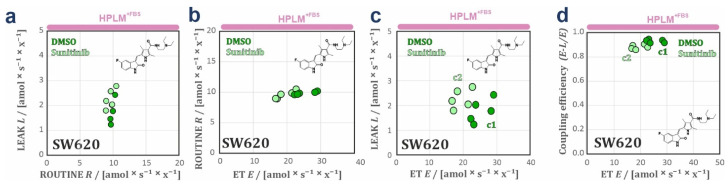
Bioenergetic cluster analyses in different respiratory coupling states of the sunitinib-exposed SW620 cells. Plots of *L* over *R*, *R* over *E*, and *L* over *E*. (**a**–**c**) SW620 cells grown in HPLM^+FBS^ and exposed to either DMSO or 10 µM sunitinib for 3 h. In this case, two heterolinear clusters (c1, DMSO and c2, sunitinib) were detected when *L* is plotted over *E*, one with higher *L* when *E* is decreasing (c2) indicating dyscoupling induced by sunitinib. (**c**) Coupling efficiencies (*E*-*L*)/*E* over ET capacities in SW620 cells grown HPLM^+FBS^ (**d**), and exposed to either DMSO or 10 µM sunitinib for 3 h. The two clusters, one with lower (*E*-*L*)/*E* values in cells grown in HPLM^+FBS^ and exposed to sunitinib (c2) indicates a drop in mitochondrial quality with the sunitinib treatment. Sunitinib chemical structure is depicted for drug-exposed SW620 cells (**a**–**d**).

## Data Availability

All data are included in the article and/or supporting information and can be available upon request.

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
