# Peer review of "Physiological Cell Culture Media Tune Mitochondrial Bioenergetics and Drug Sensitivity in Cancer Cell Models"

_cancers, 2022, doi:10.3390/cancers14163917_

Round 1

Reviewer 1 Report

In this article, the authors described the precise impact of cell culture media compositions on cell proliferation and mitochondrial bioenergetic profiles of several standard cancer cell lines and evaluated  whether HPLM influences mitochondrial drug sensitivity in cells exposed to the sunitinib drug.

The study is well conceived and executed. The article is written clearly and in a logical sequence. The experimental results and the relevant findings are presented with accurate interpretation and with a clear description. On the basis of these considerations, this reviewer recommends the publication after the following minor revisions:

·         Figure 3 g-h-i.  Please define the ratio E/E. Nowhere is explained in the text.

·         The discussion should be implemented by the explanation about the correlation between cell proliferation kinetics and mitochondrial bioenergetics. They say: In SW620 cells, respiratory rates were not affected but changes in coupling-control ratios related to ROUTINE respiration indicate that HPLM influences energy demand which correlates with lower proliferation rate in HPLM. But why the MCF 7, show increase of ROUTINE respiration, ET capacity and higher coupling efficiency, but not higher proliferation rate?  Please, discuss.

Reviewer 2 Report

Torres-Quesada et al. reported the utility of a new human plasma-like media (HPLM) in mitochondrial metabolism that affected drug sensitivity. The authors estimated the effects of MPLM on cancer cell lines in mitochondrial respiration with high-resolution respirometry (HRR) and cell proliferation. The estimation of mitochondrial respiration in new media is important. This manuscript was well-written and interesting. However, I have several concerns.

 Major concerns

  1. The authors did not make clear why they selected the three cell lines SW620, MCF7, and A375. This should be made clear.
  2. HPLM is a commercial medium, and the component is already public. Did the authors re-estimate all mediums with dialyzed FBS? If not, Figure 1 should be a supplementary figure or table.
  3. Why did the authors use sunitinib? Sunitinib is not a standard treatment for colon cancer (the origin of SW620), breast cancer (MCF7), and malignant melanoma (A375).
  4. Because the authors mention drug sensitivity,  the ratio of apoptosis to survival cell number with sunitinib treatments should be shown.

Minor concerns

  1. Though the “U-test” is more common in non-parametric tests, the authors should nonetheless consider the parametric or non-parametric in the cell line experiments.
  2. The order of Figures 4 and 5 is confusing. Figure 5a-I should be Figure 4a-I, Figure 4 should be Figure 5, and Figure 5J-m should remain, and new figure 5 should focus on the effect of sunitinib.

Round 2

Reviewer 2 Report

My all concerns were well addressed.